# Combating the COVID-19 Epidemic: Experiences from Vietnam

**DOI:** 10.3390/ijerph17093125

**Published:** 2020-04-30

**Authors:** Bui Thi Thu Ha, La Ngoc Quang, Tolib Mirzoev, Nguyen Trong Tai, Pham Quang Thai, Phung Cong Dinh

**Affiliations:** 1Department of Reproductive Health, Hanoi University of Public Health, Hanoi 100000, Vietnam; bth@huph.edu.vn; 2Department of Epidemiology, Hanoi University of Public Health, Hanoi 100000, Vietnam; 3Nuffield Centre for International Health & Development, Leeds Institute of Health Sciences, University of Leeds, Leeds LS2 9NL, UK; T.Mirzoev@leeds.ac.uk; 4School of Preventive Medicine and Public Health, Hanoi Medical University, Hanoi 100000, Vietnam; trongtai@hmu.edu.vn; 5Communicable Disease Control Department, National Institute of Hygiene and Epidemiology, Hanoi 100000, Vietnam; phamquangthai@gmail.com; 6National Agency for Science and Technology Information, Ministry of Science and Technology, Hanoi 100000, Vietnam; dinhpc.it@gmail.com

**Keywords:** COVID-19, epidemic, pandemic, prevention and control measures, public health

## Abstract

The COVID-19 pandemic is spreading fast globally. Vietnam’s strict containment measures have significantly reduced the spread of the epidemic in the country. This was achieved through the use of emergency control measures in the epidemic areas and integration of resources from multiple sectors including health, mass media, transportation, education, public affairs, and defense. This paper reviews and shares specific measures for successful prevention and control of COVID-19 in Vietnam, which could provide useful learning for other countries.

## 1. Introduction

On 31 December 2019, the World Health Organization (WHO) China Country Office was informed of cases of pneumonia with unknown etiology or cause, detected in Wuhan City, Hubei Province of China [1]. Early in the outbreak, the virus was called 2019-nCoV but later was given its official name COVID-19 [2].

According to the WHO report dated 3 April 2020, the coronavirus COVID-19 has affected 206 countries and territories, with a total of 972,303 confirmed cases and 50,322 deaths. The cases started to climb in the South-East Asia Region with a total of 5881 confirmed cases and 245 deaths, and there is a need for a stronger whole-of-society approach [3].

The first case of COVID-19 in Vietnam was declared on 23 January 2020, when a 65-year-old Chinese man became ill with fever on 17 January, four days after he and his wife flew in to Hanoi from the Wuchang District in Wuhan [4,5]. On 4 April 2020, there were a total of 240 confirmed cases with no deaths, and Vietnam was ranked 94th in the list of 206 countries and territories affected by COVID-19. The youngest affected individual was a 3-month-old female, and the oldest was an 88-year-old female [6].

In Vietnam, to the end of March, the response to the COVID-19 outbreak involved three stages. The first stage was from 23 January to 13 February, when all 16 recorded cases had completely recovered. Stage two was from 6 March to 21 March, with 78 positive cases, when the patients’ close contacts were closely monitored, which included hundreds of people. Phase three began on 22 March, after the number of infected cases exceeded 100, a threshold after which disease control becomes more challenging [7] and cases without clearer sources of infection started to appear.

Multiples effective measures have been implemented since the beginning of January 2020 and Vietnam continues to implement strict measures to contain the spread of cases. The most recent Directive 16 on strict social distancing from 1–15 April 2020, reflects the highest control of COVID-19 [8]. These measures resulted in the low spread of cases and no deaths reported by 4 April 2020. This paper aims to share these preliminary experiences of the Vietnamese Government to inform efforts by other countries in containing the spread of COVID-19. A caveat is appropriate here, which is that the purpose of the paper is not to evaluate effectiveness of government measures or indeed clinical outcomes, but to focus on documenting the experience in setting measures to control the spread of COVID-19.

## 2. Methods

We report results from a rapid review of policy documentation in Vietnam. These were obtained through the analysis of a database of recent policies, official press, articles, reports, briefs, presentations, and credible data sources in Vietnam. A total of 21 documents were obtained during the period between early January and 10 April 2020, for analysis. A thematic approach to analysis was used to identify the emerging lessons, which then informed the structure of the reported results.

## 3. Results

### 3.1. Rapid Response

A week after the first case, on 30 January, the Taskforce Group on COVID-19 prevention and control was formed with 24 members from 23 ministries, committees, the press, and radio and television representatives, to direct and coordinate among the ministries, ministerial-level agencies, government-attached agencies, concerned agencies and localities, in the prevention and control of acute respiratory infections caused by new strains of the virus. The Taskforce Group was under the leadership of Vice Prime Minister Vu Duc Dam [9]. Since the establishment of the Taskforce Group, several directives and decisions on strengthening measures to prevent and combat COVID-19 were issued with key directions: to suspend flight authorization for all flights from Chinese and other epidemic areas to Vietnam and vice versa; to limit crowds of people, especially at festivals; to suspend festivals which have not yet been opened, and to reduce the scale of the festivals held; to ask people to wear masks in public places; and to limit spring travel and participation in festivals [10].

### 3.2. Clear Leadership

The Prime Minister made several key messages towards COVID-19, such as fighting the epidemic is like fighting an enemy, and the government is willing to sacrifice economic benefits in the short term for the health of the people and work with principles that do not leave anyone behind [11]. These messages invoked the whole nation to be ready to combat the COVID-19. Figure 1 shows the multiple measures taken by different stakeholders, sectors, and levels over the country to combat COVID-19 under the direction of the Taskforce Group.

### 3.3. Supported Clinical Care and Emergency Public Health Response

Among these stakeholders, the Ministry of Health played a critical role. The Emergency Public Health Operations Center was activated in the General Department of Preventive Medicine to guide the provincial Center for Diseases Control (CDCs) on how to prevent and control COVID-19 through epidemiological actions: detection of cases, isolation, tracing cases, and the surveillance of suspected cases and close contact groups. At the beginning of the COVID-19 epidemic, Vietnam took the initiative to apply early measures, which went beyond the recommendations of the World Health Organization. Vietnam was the first country in the world to apply medical declarations. It became compulsory for everyone entering from China from 25 January 2020, and for all people arriving from elsewhere from 7 March 2020. Vietnam is also one of the few countries to apply measures to suspend visa exemption and restrict entry. In particular, Vietnam applied concentrated isolation with people who entered from or through epidemic areas and with everyone who entered from 21 March 2020. Information about disease prevention, confirmed case reports, and exposure findings were broadcasted daily by almost all channels of the national Vietnamese media. National phone ringtones were also changed to give notice of COVID-19. People were required to install NCOVI health notification software and report daily the health status of themselves and their family members. During the period from 1 April to 15 April 2020, Vietnam implemented social isolation measures across the whole country to prevent the spread of the epidemic in the community.

The admission of medical services set up the center for management of clinical support for COVID-19 patients. Despite the lack of specific treatment, the center issued several guidelines on the medical treatment of COVID-19 patients that were adapted to the Vietnamese context [12,13]. and implemented in all hospitals designated for COVID-19 treatment. The first clinical guideline on diagnosis and treatment of COVID-19 patients according to clinical manifestation was issued in February [12]. Later, the guideline was adjusted according to upgraded global scientific reports on COVID-19 treatment in March [13]. These guided hospitals that treated COVID-19 patients.

Local commune health centers provide primary health care services in Vietnam. Their frontline staff played critical roles in the prevention of COVID-19. They provided health education on preventive measures for all people in the local areas, identified people who had been in close contact with COVID-19 patients or those coming back from epidemic areas, and guided them through the procedures: identifying local isolation points or home isolation with medical checks, such as temperature records or health check-up, and being sent for referral if necessary [14].

The Ministry of Health (MOH) also organized several site visits to support local health facilities to prepare for combating COVID-19 and mobilized provision of medical equipment and protective measures for health facility workers

### 3.4. Multi-Sectoral Approach

Other sectors joined as needed under the guidance of the Taskforce Group (Figure 1). The media communicated closely with the Ministry of Health (MOH) on disseminating information on the prevention and control of COVID-19 from the beginning. Official newspapers, government’s website [15], MOH’s website [16] and open TV channels provided daily updates on positive cases globally and in Vietnam, and conveyed MOH health messages to prevent and control COVID-19, to large audiences [16]. The broadcasting of specific new cases on national TV, and their related epidemiological information, allowed high risk groups to be traced all over the country, especially during 7–20 March, when many citizens returned to Vietnam from Europe and the United States. The repeated communication on the prevention of COVID-19 (wearing masks, hand washing), and the promotion of social distancing (stay home and keep distance from others at 2 m minimum), was helpful in changing people’s behavior toward the epidemic more seriously. Transparent information on positive cases helped to convey the image of government action towards COVID-19. Furthermore, the Ministry of Health created an official account on social media (Zalo), sent SMS to all citizens, and changed waiting ringtones to a voice message to remind about COVID-19 protection measures.

### 3.5. Outstanding Challenges

Despite the preliminary success in slowing the spread of cases in Vietnam, several challenges are ahead. The lack of medical equipment (ventilators in ICU) and lack of personal protective equipment (masks and gowns) for medical staff in the hospitals have been reported [17,18]. The government is looking for supplies from different sources to ensure the sufficient numbers for hospitals are obtained.

The decision for the compulsory isolation of people returning from abroad for 14 days in local centralized isolation points came into effect only on 20 March 2020, which left a large window of 1–20 March 2020, during which many unknown imported positive cases [19] may have been transmitted within the community.

The public is flustered and still lacks awareness, which has informed a lot of rumors and fake news related to COVID-19 in the early period and has fueled unnecessary action of the community, such as panic buying of food and other goods. Some groups have ignored social distancing measures and have still met in bars or even have held banquets [20], which has created potential sources for the transmission of cases.

Although Vietnam has contained its COVID-19 cases up to now, the government is planning to apply stricter measures to prevent COVID-19 in the longer-term through the introduction of compulsory mask wearing and enforcing the implementation of social distancing with standing at least 2 m apart in public and avoiding gatherings in large numbers; to enhance early detection, testing cases with flu-like symptoms, unexplained deaths, and pneumonia cases visiting medical facilities; to identify contact cases; to continue to restrict entry from other countries, and to carry out the detection, isolation, and quarantine in the country; to apply concentrated isolation for 14 days for people who have entered Vietnam; to continue enhance communication to raise awareness and to implement measures to prevent and control the COVID-19 epidemic for individuals, families, and communities; and to apply new research in active treatment and improving survival rates.

## 4. Discussion

The COVID-19 pandemic poses a significant threat to global health and is a big challenge for all countries. National governments need to understand the effective measures to prevent the spread of cases. The on-going experience from Vietnam highlights the practice of the introduction of early diagnosis, prevention, and treatment guidelines.

A strong leadership and commitment from the highest political level (in Vietnam’s case, the Prime Minister) was another important influence, complemented by the coordinated joint efforts of the Taskforce Group. This helped to mobilize different sectors and levels to engage in the prevention and control of COVID-19.

The continuing efforts of the health system (CDC network and hospitals) in the control, prevention, and curative care of COVID-19 are other key effective measures, reflected in their visibility and clear action on detection, confirmation of cases and suspected close contacts, and intensive care for positive cases. Moreover, the joint efforts of different sectors and levels in tracing suspected and close contacts, the strict isolation of cases, and having local isolation points for people returning abroad for 14 days, help to minimize the risk of transmission in the community.

The transparency of updated information and clear communication messages on COVID-19 through official and social media were important contributors to changing community behaviors towards wearing masks, hand washing, and social distancing, from February 2020. Before closing, we would like to reflect on the wider applicability of the above experiences and practice. Vietnam has long borders with China and still has contained cases without deaths due to COVID-19. These above-mentioned experiences and practice could be applied to other ASEAN countries that have similar socio-political characteristics, such as Laos and Cambodia. We recognize, however, there are different approaches to containing COVID-19. For example, the South Korean government applied a far more liberal approach with no lockdowns, underpinned by mass testing [21], which is conceptually different to the strict measures deployed in Vietnam. An appropriate combination of the government’s command, control, incentives, and communication is a key to ensure the public’s compliance with the government agenda. In Vietnam, a stronger emphasis was made on the central management and leadership by the MOH and the government. However, in other contexts with more diverse political environments this approach may need adapting, for example, in the multi-party political context perhaps more liberal approaches may be more appropriate. The introduction of reporting of personal and family health status needs to consider issues of individual privacy and data confidentiality, which are often a source of large public discourse, for example, in European contexts. Ultimately, any responses in this unprecedented situation need to be globally and nationally context-specific, and ideally informed by documented experiences and practice from different contexts.

## 5. Conclusions

In this paper, we have summarized the on-going experience in reducing the spread of COVID-19 in Vietnam. The Vietnamese response is characterized by rapid response, clear leadership, a multi-sectoral approach, and supported by clinical care and a public health response. The wider applicability of these experiences is subject to differences in socio-political environments, which determine public compliance with the government agenda. We did not systematically evaluate effectiveness of these measures, which represents a possible area for future research on this topic.

## Figures and Tables

**Figure 1 ijerph-17-03125-f001:**
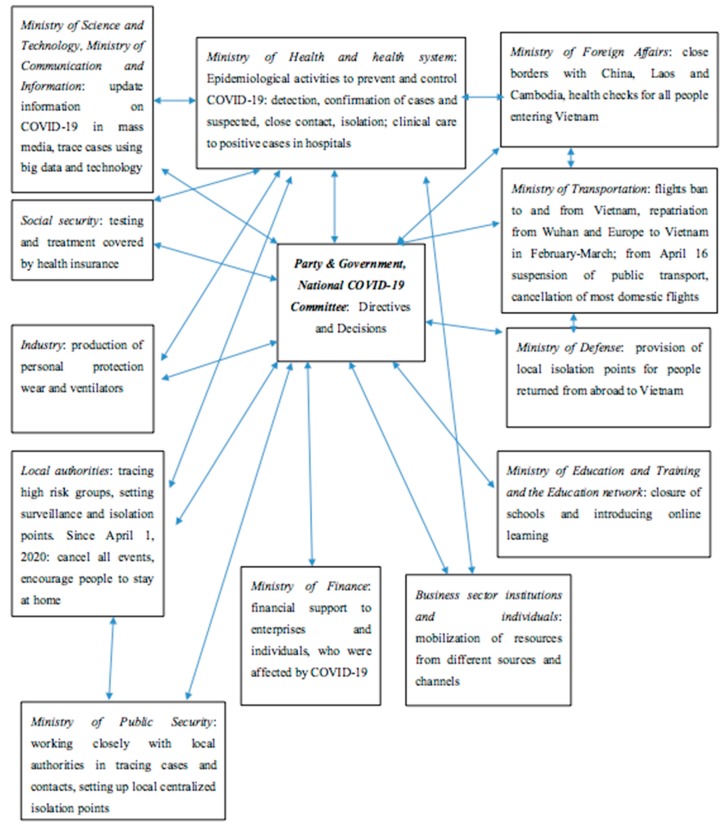
The coordinated national response to COVID-19 in Vietnam.

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
