# Peer review of "Combating the COVID-19 Epidemic: Experiences from Vietnam"

_ijerph, 2020, doi:10.3390/ijerph17093125_

Round 1
Reviewer 1 Report
Dear Authors,
I appreciate your manuscript on combating COVID-19 in Vietnam, however i have few concerns which need to be addressed;
- would you add key points related to treatment plans in Vietnam?
- key points on preventive measure taken so far?
- Next steps for combating the COVID-19 spread in Vietnam?
- add notes on frontline and other facilities in Vietnam?
Thanks,
Reviewer 2 Report
Thank you very much for allowing me to review the article “Combating COVID-19 epidemic: the Vietnamese experience and implication for other countrie” (ijerph-780392).
This work focuses on the COVID-19 pandemic, specifically in Vietnam. In order to better understand how Vietnam has managed to reduce the public health impacts of the COVID-19 pandemic, this paper reviews the specific measures for prevention and control of the COVID-19, which could be replicated in other countries.
Comments:
Summary: it must have methodology, results and conclusions. The abstract should allow the reader to get an idea of the complete paper, not just its intention.
The introduction is highly structured and informative. At the end of the introduction the objective of the study should be indicated.
"2. Responses to the COVID-19”, is the section that explains the measures that were adapted, but this section in a scientific journal corresponds to“ material and methods ”. And in this section it should be explained that based on the measures adopted, the effect will be assessed ... The measures are taken at the government level, the impact is assessed in the work, they should not be explained as if they were taken by the authors of the work.
No results.
There is no discussion.
This is a descriptive study that shows the preventive measures adopted by the Vietnam´s government in the face of the COVID-19 pandemic, but they are not valued. Nor does it consider that the affected countries are very different from the population, social and, of course, health system perspective in each one. These differences mean that the measures adopted in one country are not directly applicable to other countries, it is necessary to adapt them to the peculiarities of each country.
Round 2
Reviewer 2 Report
I have carefully reviewed the article "article" Combating COVID-19 epidemic: the Vietnamese experience and implication for other countries "(ijerph-780392)." and the responses of their authors.
This is an article that describes the measures taken in Vietnam, it does not evaluate its effects on COVID-19 epidemic.
It is well written, informative and will serve as the basis for future studies.
Its authors have completed important aspects and have taken into account the limitations of the work.
It allows us to know the measures adopted and the speed of action and serve for other sociocultural similar communities.
Comments.
The authors have to understand that their work is descriptive, they cannot interpret the measures adopted, future works may assess its effectiveness.
In the discussion, line 155 delete the phrase "The on-going experience from Vietnam highlights a number of useful lessons". this phrase is very speculative. Delete the sentences on line 158 and 159, this is a scientific work so there should be no political aspects, it supposes a scientific conflict of interest.
In the discussion, the measures adopted by other countries should be evaluated for similarities and differences with those adopted in Vietnam.
Conclusion: Adjusted to the fact that it is a descriptive study, you cannot speak of giving lessons, you have not seen its effects, they are only suggestions.
Author Response
Response to Reviewer 2 Comments
We would like to say thank you very much for your comments and suggestions for our article. There are some explanations according to your comments and suggestions:
This is an article that describes the measures taken in Vietnam, it does not evaluate its effects on COVID-19 epidemic. It is well written, informative and will serve as the basis for future studies. Its authors have completed important aspects and have taken into account the limitations of the work. It allows us to know the measures adopted and the speed of action and serve for other sociocultural similar communities.
Comments.
The authors have to understand that their work is descriptive, they cannot interpret the measures adopted, future works may assess its effectiveness. In the discussion, line 155, we take this point and while we have retained the contents we made it clear that we are not reporting these issues as important lessons learned
We reported as practice: “The on-going experience from Vietnam highlights the practice of introduction of early diagnosis, prevention, and treatment guidelines”
Delete the sentences on line 158 and 159, this is a scientific work so there should be no political aspects, it supposes a scientific conflict of interest.
We delete the sentences on line 158 and 159.
In the discussion, the measures adopted by other countries should be evaluated for similarities and differences with those adopted in Vietnam.
As recommended, we have now included discussion of similarities and differences of South Korean experience to Vietnamese.
Conclusion: Adjusted to the fact that it is a descriptive study, you cannot speak of giving lessons, you have not seen its effects, they are only suggestions.
We understood this limitation and have changed the word from lesson into experience and practice.
(Please see the attachment as well)
